# Patients' experiences of the complex trust-building process within digital cardiac rehabilitation

Marjo-Riitta Anttila[1]*, Anne Söderlund[2], Tuulikki Sjögren[1]

**1** Faculty of Sport and Health Sciences, University of Jyväskylä, Jyväskylä, Finland, **2** School of Health, Care and Social Welfare, Mälardalen University, Västerås, Sweden

☉ These authors contributed equally to this work.

* marjo-riitta.m-r.anttila@jyu.fi

## Abstract

The development of digital solutions is becoming increasingly important in facing global challenges. Therefore, research on this topic is important in taking into account cardiac patients' experiences of the rehabilitation process for the design of digital counseling solutions. The aim of the present qualitative study was to explore the different meanings that patients give to the rehabilitation process using a Glaserian grounded theory (GT) approach. Qualitative interviews were conducted with 30 participants from a rehabilitation center in Finland. The findings indicated a "complex trust-building process" core category comprising five categories of trust-building in rehabilitation: feeling that one has hit rock bottom, facing and coping in a crosscurrent, understanding together as a peer group, moving toward a healthier lifestyle with technology, and finding self-awareness. The complex process of trust-building involved interactions among emotion, cognition, and acceptance and support processes. Therefore, digital rehabilitation should be incorporated into counseling based on patients' psychosocial, physical and emotional needs to help patients become aware of their own feelings and thoughts during the rehabilitation process.

## Introduction

The development of digital solutions needs to be increasingly taken into account in health care [1] and cardiac rehabilitation [2,3]. Cardiac patients have also shown increasing interest in using healthcare technology [4–6] and participating in remote rehabilitation due to increasing digitization [7]. Cardiac patients are attracted to the ability of remote rehabilitation to provide convenient and flexible access to real-time personalized support [4,7]. Cardiac patients' experiences of lifestyle change issues in the digital context, such as behavior [8–10], physical activity [11], healthy eating [12], social support [13–15] and mental health [16–18], have been studied in the context of common remote interventions. However, none of these studies examined rehabilitation as a whole process.

Notably, after heart events, cardiac patients experience anxiety and depression [19,20], impairment of cognitive function and emotional distress [21], lack of social support [20] and

**Data Availability Statement:** Our research is qualitative research data collected (The study is a part of a remote technology in cardiac rehabilitation study registered at the ISRCTN Registry [ISRCTN61225589]). As authors, we have not

placed excerpts from transcripts relevant to the study in a suitable repository because sharing the excerpts violates the consent of the participants. Upon request, we may share analytics data that does not contain identification data. Data requests may be sent to The Ethics Committee of the Central Finland Health Care District The Ethics Committee of the Central Finland Health Care District Päivi Lampinen paivi.lampinen@ksshp.fi Keskussairaalantie 19, rak. 1, Minikampus (2. krs) 40620 Jyväskylä, Finland puh.+35814 269 5134.

**Funding:** University of Jyväskylä and Social insurance institution of Finland (Kela) The University of Jyväskylä is employer by researchers, Tuulikki Sjögren and Marjo-Riitta Anttila (salary). The authors received no specific funding for this work (article). The funders (Kela) had no role (no salary) in decision to publish, or preparation of the manuscript by any authors.

**Competing interests:** The authors have declared that no competing interests exist.

impacts on their self-care management. Therefore, an important aim of the cardiac rehabilitation process is to support patients' ability to adapt. There are a number of theoretical frameworks to explain the adaptation process after a cardiac event, such as the biomedical, psychological and biopsychosocial models of adaptation to illness [22,23]. The process of adjustment or adaptation to chronic illness refers to an individual's psychological reactions or responses to an illness or disability and their reorganization of their lives due to the condition [24]. These reactions also involve the meaning-making process, by which an individual continues to search for new situational meaning and relate this meaning to his or her general ways of understanding life and the world [25]. Theoretical perspectives, such as cognitive behavioral theories and treatment approaches [26–28] and social cognitive theory (SCT) [29], are helpful for understanding how an individual's behavior develops and changes as part of the rehabilitation process. In the rehabilitation process, it is important to take into account the interaction between an individual's thoughts and feelings, past experiences and behaviors [27], as well as the social environment [29]. An individual's acceptance and awareness as well as his or her engagement and behavior change strategies increase his or her psychological flexibility in the behavior change process [27]. In addition, in the rehabilitation process, the promotion of the individual's positive thoughts and beliefs alleviates emotional distress and reinforces self-esteem, which in turn contributes to external changes in behavior [27,29].

Patients' own rehabilitation processes are supported by their coping strategies and ability to adapt cognitively to the disease [24], and community social support is an important part of this process [29,30]. Coping strategies are the means that cardiac patients use in their cognitive, emotional and behavioral efforts to manage or adapt to the demands of the cardiac event that exceed their resources [31]. Coping mechanisms are often divided into two groups: problem-focused and emotional-focused coping strategies [31]. Furthermore, coping may be differentiated into approach behavior and avoidance behavior, which refer to, e.g., a patient talking about emotions with his or her peer group (approach) or trying to forget the cardiac event (avoidance) [32]. In the rehabilitation process, peer support is important and is facilitated by peers' ability to share experiences of illness or disability [30]. Cardiac patients' sense of connection with other cardiac patients as part of the rehabilitation process can decrease isolation and increase psychological well-being [20].

Notably, numerous qualitative studies have described patients' perceptions of the cardiac rehabilitation process after a cardiac event [33–37]. Qualitative studies have gradually begun to examine the experiences of patients with digital programs [7,38,39], but the present qualitative study is even more broadly related to the multidisciplinary digital cardiac rehabilitation process than previous studies. This research goal is important because there is a need to expand the understanding of the experiences of the rehabilitation processes of patients who have used digital technology. The aim of the present qualitative study was to explore the different meanings patients give to the rehabilitation process using a grounded theory (GT) approach.

## Methods

The study analyzed patients' experiences of the rehabilitation process by using the Glaserian GT method [40]. We chose to use a GT approach because it is suitable for investigating phenomena to develop and apply knowledge about patients' concerns regarding social interaction in digital rehabilitation. In GT, the research problem is called the main concern, and it explains the primary focus of the substantive area of research [40]. An inductive methodology was used in the analysis process, which allowed us to approach the research with a completely open mind and without any preconceptions of the results. The analysis process required theoretical sensitivity for the research results to support the conceptualization of the data [41].

## Setting

Semistructured interviews were conducted with cardiac patients (mean ages 55.46), including 22 males and 8 females, who had undergone coronary angioplasty (PTCA) (25/30) or coronary artery bypass graft (CABG) (4/30) approximately 3 to 12 months prior to rehabilitation. In Table 1 demographic characteristic of the participants at baseline is presented. The participants were recruited among adult patients with coronary artery disease (CAD) and attended the cardiac rehabilitation courses that utilized digital health tools for traditional cardiac rehabilitation (Peurunka rehabilitation centre, Finland). The inclusion criteria for participation were adults (18 years or older) who had cardiovascular risk factors or angina pectoris with physical working capacity limitation or myocardial infarction or coronary artery bypass graft surgery (CABG) or coronary angioplasty (PTCA). Participants were excluded if they had musculoskeletal problems, cognitive or memory impairment, or if they were unable to use independently computer and remote technology application. Participation in the study intervention also required the participant to have internet at home or be able to use a computer of family members or friends. Participants in the study received an activity tracker accelerometer (Fitbit Charge HR®, USA) and remote coaching tool software (m-coach Movendos, Finland), which

**Table 1. The demographic characteristic of the participants at baseline.**

|  | Percent % | n/total |
|---|---|---|
| **Gender** |  |  |
| Men | 73.3 | 22/30 |
| Women | 26.7 | 8/30 |
| **Marital status** |  |  |
| Married | 60.7 | 17/28 |
| Unmarried | 7.1 | 2/28 |
| Cohabiting | 10.7 | 3/28 |
| Divorced/separated | 17.9 | 5/28 |
| Widowed | 3.6 | 1/28 |
| **Education** |  |  |
| Vocational or course-form school or other | 28.6 | 8/28 |
| College-level education | 42.9 | 12/28 |
| University of applied sciences | 25 | 7/28 |
| University | 3.6 | 1/28 |
| **Employment status** |  |  |
| Retired | 28.6 | 8/28 |
| On sick leave | 3.6 | 1/28 |
| Unemployed or laid off | 3.6 | 1/28 |
| Working full-time | 53.6 | 15/28 |
| Working part-time/part-time retiree | 3.6 | 1/28 |
| Others | 6.7 | 2/28 |
| **Operation** |  |  |
| Coronary artery bypass graft (CABG) | 13.3 | 4/30 |
| Coronary angioplasty (PTCA) | 83.3 | 25/30 |
| No operation | 3.3 | 1/30 |
| **Time of operation** |  |  |
| 0–3 months from rehabilitation | 6.7 | 2/30 |
| 3–6 months from rehabilitation | 16.7 | 5/30 |
| 6–12 months from rehabilitation | 46.7 | 14/30 |
| Over 12 months from rehabilitation | 26.7 | 8/30 |

was free to download on computers, tablets and smart phones. Patients were informed verbal and written of the study prior to rehabilitation that participation was voluntary and that they could withdraw from the study at any time. Patients also signed informed consent forms to participate in the study. Patients were also assured that whether they wanted to participate in the study or not, it would not have any negative impact on the rehabilitation process.

The rehabilitation intervention lasted 12 months and included three five-day conventional rehabilitation periods at a rehabilitation center offered by the Social Insurance Institution of Finland. The rehabilitation course provided guidance on various topics related to a changed life situation due to cardiovascular disease. The course involved exercise and group-based educational and discussion sessions given by a doctor, a physiotherapist and a nurse, and optionally, a social worker, a psychologist and a dietitian. Between the rehabilitation periods, the intervention provided patients with web-based coaching via a remote connection using web-based software (m-coach Movendos, Finland) and an activity tracker accelerometer (Fitbit Charge HR®, USA). The web-based coaching consisted of monthly tasks that aimed to increase patients' ability to cope with everyday life with cardiac illness. The physiotherapist contacted patients monthly using the software and provided feedback on their rehabilitation process. In addition, the web-based software also sent automatic motivational messages every month. The coaching software allowed patients to share their own experiences with the peer group and to send a message to their own physiotherapist if needed.

### Data collection and analysis

Data collection was guided by a purposeful sampling strategy, which involves the purposeful selection of data samples according to the developing categories and emerging theory [41]. This sampling strategy is called theoretical sampling in the GT method, whereby the researcher concurrently collects codes and analyses data and decides during the analysis process which data will be collected next [41]. Following this strategy, we analyzed the cardiac patients' interview material at the beginning of rehabilitation (0 months) [4], and the analysis results obtained therein guided data collection at the end of rehabilitation (12 months).

The patients were divided into four focus groups, each of which participated in a semistructured interview conducted at the rehabilitation center at the end of rehabilitation in 2016 and 2017. The study is a part of a distance technology in cardiac rehabilitation study registered at the ISRCTN Registry [ISRCTN61225589]. The Ethics Committee of the Central Finland Health Care District approved the study. Each focus group interview included 5–9 patients and lasted an average of 30–90 minutes. The participants were encouraged to speak openly about their experiences of the rehabilitation process. The interviewer was free to vary the questions and their content and to allow for group discussion. Example questions, are as follows: "What were your meaningful experiences with rehabilitation process?"; "How was the peer group meaningful to you?"; and "What are your experiences of how technology has been a help to you in coping with daily life and lifestyle changes?" The interviewer asked follow-up questions as needed. The interviewer observed the group dynamics and ensured that everyone had an opportunity to speak. Occasionally, there were people in the focus group who talked more and guided the session, but the interviewer provided a balanced discussion by asking the quieter participants supplementary questions. Interviews were tape-recorded and transcribed with ATLAS.ti software (ATLAS.ti Scientific Software Development GmbH). In addition, manual techniques, such as systematic "sentence by sentence" analysis, were used to facilitate data analysis [40]. Throughout the constant comparative process of data collection and analysis, free-form memos were written, and interpretations were discussed in the research group according to the GT approach [40,42].

The first step of the data analysis process was "open coding", during which we were open to all interpretations [40]. We became familiar with the data by carefully and repeatedly listening to the recorded interviews. We reread the interview transcripts to familiarize ourselves with "what was going on in the data" and the "main concern" of the participants [40,42]. In the second step of data analysis, we constantly compared incidents to incidents, incidents to concepts, and concepts to concepts [40,41]. Then, we categorized the codes to form subcategories and categories to conceptualize the experiences of patients. As we created the categories, we looked for connections among the codes and tried to understand how one code related to another. We analyzed whether one concept should be categorized with another concept or whether each concept should become its own category. We also looked for important descriptive words and meaningful expressions that participants used. These in vivo terms, which were one or two words in length, described what was happening in the data [40].

In the third step, the analysis process continued until a coded category and its properties became saturated by constant comparison [40]. Open coding was complete when the core category was identified. In the fourth step, the data analysis process continued with the selection of a core category and the determination of the properties of the categories of the core category [40]. We returned to the raw data and checked our analysis by recoding and comparing the new categories to those created while writing memos. We examined whether these categories and their properties fit into the phenomenon that was becoming the core category [40]. Fig 1 presents the systematic data collection and analysis process.

## Results

The experience of rehabilitation was described as a complex trust-building process by cardiac patients in digital rehabilitation. This process reflected the emotion, cognition and acceptance and support processes that patients experienced during rehabilitation. This complex process

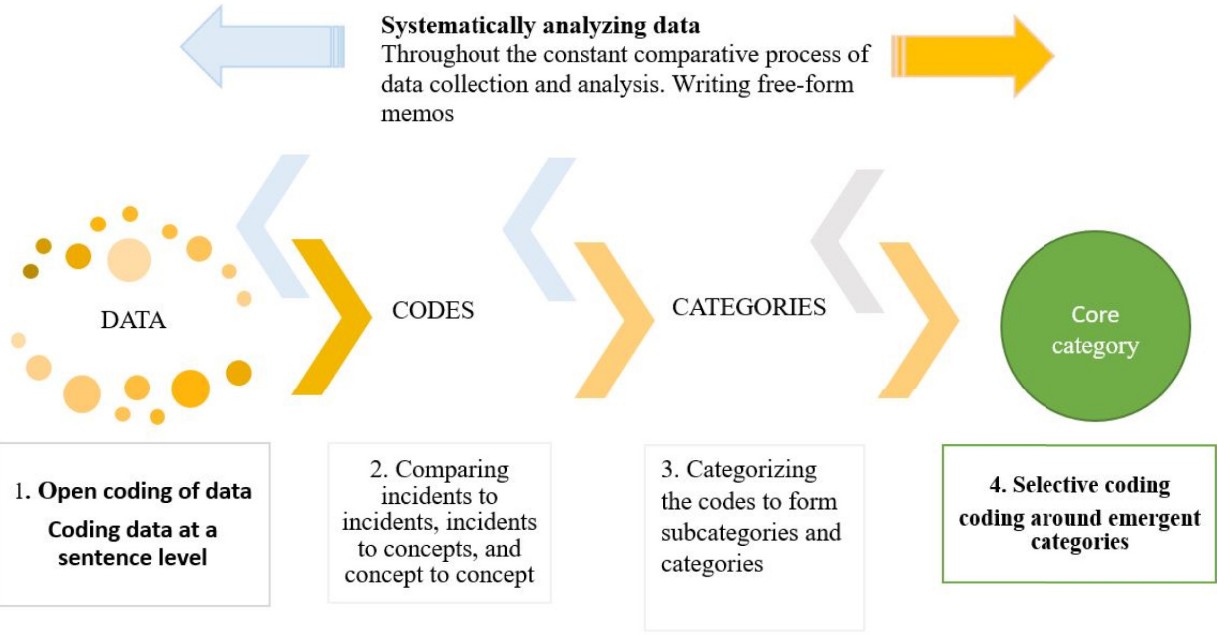

**Fig 1. Data collection and analysis process of this study.**

**Table 2. "Complex trust-building process" core category and its five categories.**

| | "Complex trust-building process" core category and its five categories | | | | |
|---|---|---|---|---|---|
| Change processes | Five categories of being one year in the rehabilitation | | | | |
| | **Feeling that one has hit rock bottom** | **Facing and coping in a crosscurrent** | **Understanding together as a peer group** | **Moving toward a healthier lifestyle with technology** | **Finding self-awareness** |
| **Emotion** | Feelings of worthlessness and powerlessness | Struggling the ups and downs | Feeling of being on the "same level" | Moving forward toward self-trust | Finding hope |
| **Cognition** | Avoiding thoughts | Noting own thoughts | Giving and receiving feedback | Seeking information about activity | Developing self-care through learning, repetition |
| **Acceptance and support** | Recognizing rock bottom of one's own life | Encouraging support | The positive importance of face-to-face peer group contact | Supporting "a great buddy" technology | A time and place to increase awareness of one's own needs |

was observed through five categories of trust-building in rehabilitation: feeling that one has hit rock bottom, facing and coping in a crosscurrent, understanding together as a peer group, moving toward a healthier lifestyle with technology, and finding self-awareness. The results obtained from the GT constant comparative analyses are summarized in Table 2.

## Feeling that one has hit rock bottom

All of our participants with cardiac illness focused on digital rehabilitation in the interviews, but a minority mentioned their feelings about the experience of cardiac illness. Patients described their feelings of distress after a heart event as the feeling of having hit "rock bottom" and recognized the heart event as the nadir of their lives. This life situation included a loss of self-esteem and negative emotions such as feelings of worthlessness and powerlessness and even despair due to their sudden experience of a heart event. They also reflected on the connection between work stress and heart events. Their involvement of the cardiac rehabilitation course quickly after the cardiac event, such as a cardiac surgery or angioplasty stent, was considered important. The participants also noted that going to rehabilitation challenged them to address their own thoughts and feelings about managing their new life situations. They tried to avoid thoughts or feelings that made them insecure. These reactions hindered their coping in daily life, which was reflected in one interviewee's description of feelings of experiencing cardiac illness:

> "So... rock bottom becomes my life... When I felt myself, like that I was nothing anymore. That's it. I thought I couldn't do anything" [Participant 18, 64-year-old woman, focus group 5].

## Facing and coping in a crosscurrent

After a heart event, some patients faced crosscurrents in life. They described struggling with difficulties in life and experiencing both the ups and downs. On the other hand, some participants faced outside pressure in pursuit of a new lifestyle. Most of them wanted to return to work, and retraining had been suggested for some patients. However, some participants felt relieved to stop working, such as due to retirement. The patients' descriptions showed that they did not want to give up and that the support they received encouraged them to continue. Patients highlighted the importance of support in facing adversity and coping in a new stage in life. They felt that friends and family members and colleagues were the most important people who made them feel supported and encouraged. Some participants appreciated the feedback and support received from healthcare professionals, such as this patient who also noted the importance of having his own hobby:

*"Motivation is pretty good, has been good all that time. For these things, and. . . family and work friends are of course with me, and then the motorcycling. When I have a hobby, then those guys hanging out with me, and. . .the staff here have been very nice. and things have gone well for me. . .and that's about the way it is now, then. . ."* [Participant 30, 53-year-old man, pilot focus group].

The following comment describes another patient's thought process:

*"That's the way it was, when I said, when I downplayed it that way. You see, I was kind of hardheaded and headstrong. Nobody tells me what to do, but well. . .then I got to the second level, then it started to get me moving"* [Participant 4, 59-year-old woman, focus group 5].

## Understanding together as a peer group

Almost everyone in the study indicated the great importance of peer group support. In their view, the most important aspect of rehabilitation was the experience of connecting with others. They described the feeling of being on the same level as other peer group members and the possibility of forming a common understanding of what had happened as meaningful. They emphasized that shared experience, unique group support and a sense of reciprocity were key to rehabilitation. Peer group activities were described as interactions in which patients received and gave feedback to each other. Several participants emphasized the positive importance of face-to-face peer group contact, when patients could talk about digital rehabilitation. They were concerned about isolation in remote rehabilitation without social support. Most participants described the positive experience of being at the "same level" as each other and the peer group relationship as helpful for initiating their own understanding of the adaptation process; for example, one interviewee described,

*"So it is.. I feel sure that these people understand just what I've experienced myself. . . and. . .and being like on same level there"* [Participant 56, 45-year-old man, focus group 3].

## Moving toward a healthier lifestyle with technology

Patients' experiences of rehabilitation utilizing a remote technology application were diverse and included both positive and negative experiences. Many patients felt that the technology supported physical activity and lifestyle changes. Some patients noted that rehabilitation with technology was part of their learning and thought processes in a new life situation. The thought process included seeking information about activity, which enabled them to identify the level of physical activity and sleep quality. Many of them felt that technology helped them recognize their own development and noted that they received immediate feedback on their own levels of physical activity. Activity monitoring motivated them to move forward toward achieving their own goals. The physiotherapist provided feedback and support through a web-based program that also helped patients seek and utilize information to evaluate their goals and progress. Patients were aware that the activity results provided by the technology were different from their own experiences of activity, but this did not interfere with the positive experience with the technology. On the other hand, participants were frustrated with the technology due to a lack or breakdown of equipment. They described the technology as friendly or "*a great buddy*", as it helped them keep track of their daily activity. For example, one interviewee said,

*"Yeah, that became a great buddy there, I followed it. . .and just those steps, I tried for like ten-twelve thousand. But now, when it got broken, yeah, I´m like. . ."* [Participant 14, 55-year-old man, focus group 3].

### Finding self-awareness

Most of the participants described the meaningfulness of rehabilitation in providing a time and place to increase awareness of one's own needs and thoughts, set goals, and evaluate and achieve them. In addition, they found hope in living and gradually forgot about the disease in their daily lives. On the other hand, some patients wanted more personalized life management and plans that took into account their life situations. Patients described their own development in self-care that occurred through their learning, repetition, and application of self-care knowledge. They found self-confidence and satisfaction and forgot about the illness, as one woman described:

*"The fact that many times, it is so much to itself, probably, that when the disease is not here right now, it is a little over there, like just around the corner. That it doesn't remind me every minute, remind you. So, it's as easy as forgetting it. And don't take it like you should always think about it"* [Participant 26, 61-year-old woman, focus group 2].

These results indicate that the sudden onset of heart disease could have significant psychological impacts and affect the daily lives, attitudes toward work and emotions of cardiac patients. In next section, therefore, we specifically discuss cardiac patients' experiences of the meaning of the trust-building process.

## Discussion

This study aimed to build an understanding of the different meanings patients gave to the rehabilitation process. The main result of the study was the identification of the core category, the "complex trust-building process", as shown in Fig 2. Five categories of meanings of the complex trust-building were observed: feeling that one has hit rock bottom, facing and coping in a crosscurrent, understanding together as a peer group, moving toward a healthier behavior through technology, and finding self-awareness. This complex process was based on patients' different experiences and their main concerns in relation to the trust-building process. The complex trust-building process included interactions between change processes such as emotion, cognition, acceptance and support processes.

The results of this study show that the main concern of the participants in the rehabilitation process was not only related to, for example, lifestyle changes in physical activity and diet. The patients also described the rehabilitation process as relating to their psychosocial and emotional needs, as seen in previous studies [16,17,19,33–37]. On the other hand, patients described digital solutions mainly as supporting behavior change. For example, when speaking about the importance of digitalization, patients focused on the support provided by applications for lifestyle change but not for reflections about the adaptation process and the management of emotions and thoughts. They were also concerned about the implementation of the peer group in a completely digital way, as the most important aspect for them was the feeling of belonging to the peer group and sharing experiences face to face. Theoretically, the findings of this study are largely consistent with the underpinnings of cognitive behavioral theory [26–28] and SCT [29], showing how patients' trust-building occurs through emotional and cognitive processes during cardiac rehabilitation.

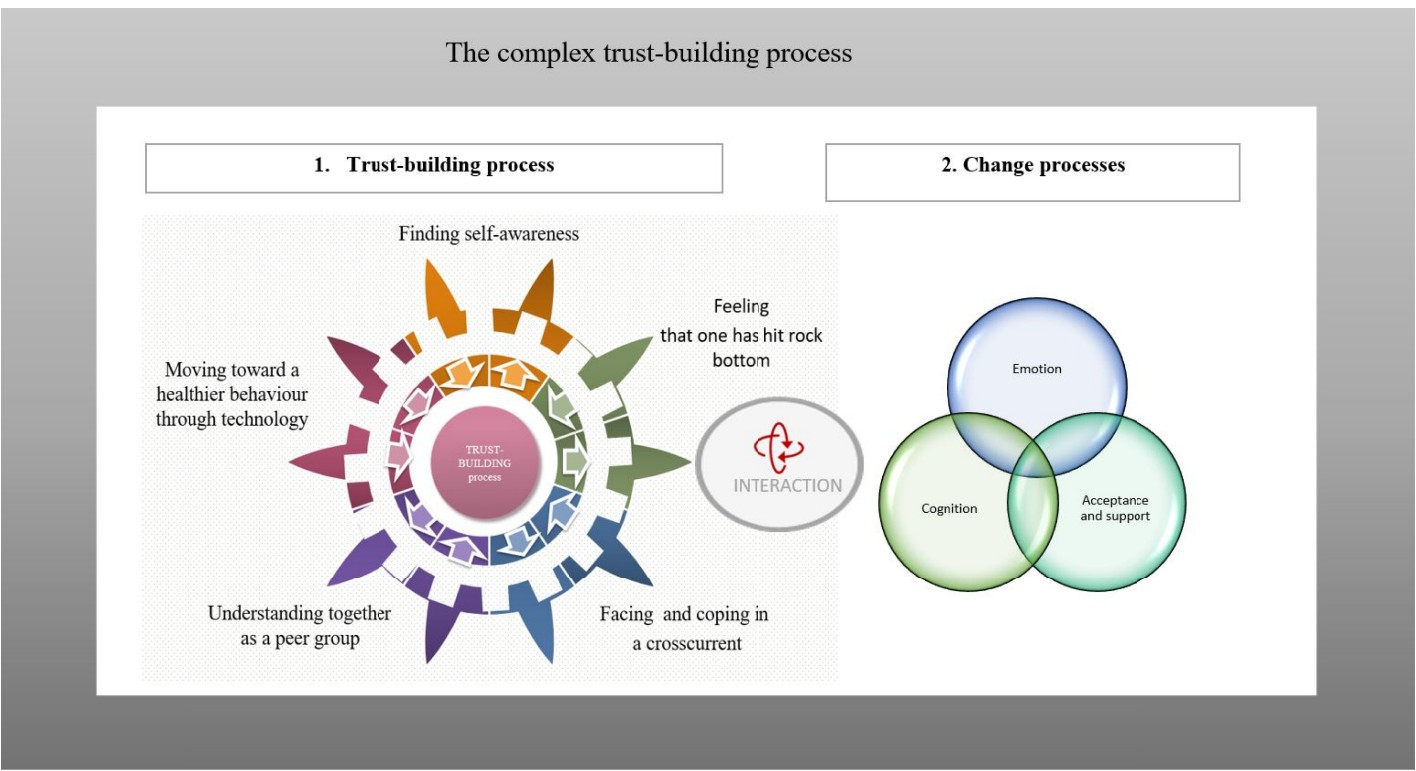

**Fig 2. Interaction between change processes in the complex trust-building process.**

## The heart event–an emotionally stressful situation

The psychological processes that patients described included feelings of worthlessness and powerlessness, as well as the loss of one's dreams, similar to "hitting rock bottom", after a heart event. Our study shows how patients understood the meaning of their own thoughts when coping in crosscurrents. It shows how these patients coped with internal (attitudes) and external (expectations to fulfill) challenges that may have affected their ability to adapt illness. This psychological reaction or response to crosscurrents also refers to the meaning-making process as an active coping tool, which was also demonstrated in a previous study [25]. Professional staff have an important task in supporting patients in identifying these emotional and thoughts processes, as has been previously stated [6,18,30], because how a person interprets and reframes a situation is relevant to how emotionally stressful a situation becomes [30].

## Supporters

Peer support and the formation of an understanding of the meaning of the disease for one's own life became especially important in social processes. Patients described the meaning of the rehabilitation peer group to them as, for example, learning together, sharing experiences and feeling a sense of belonging, and building a common understanding of the disease and coping. Such meaning arises in interactions with other peers in the same situation by giving and receiving feedback, which has also been described in another study [30]. The results of this study support those of previous studies, which have shown how coping with different stressors in life through social support buffers a stressful environment [13–15,20,30]. The current study demonstrates the role of peer groups and spouses and other family members, friends, and

health care professionals as supporters, as previous studies have shown [13,14,30]. Patients cope with emotional distress as a peer group, and identifying the significance of such coping is relevant in the trust-building process.

## Technology as part of the thinking process

In addition to the psychosocial processes described above, patients talked about the process of behavior change. Our study adds to previous knowledge by providing descriptions of how patients moved toward healthier behaviors with support from technology [8–10]. Patients reported that physical activity and sleep quality monitoring provided feedback on progress toward their goals and healthier behavior. This feedback made it possible to search for information through the monitoring of the thought process, and the use of technology as part of rehabilitation can provide patients with the opportunity to practice interactive thinking skills. [5]. These results support the findings of previous studies [7,38,39] showing that it is important for patients to also have opportunities for communication with a rehabilitation professional regarding their own health issues.

Patients desired more personalized and individualized counseling to maintain motivation during rehabilitation, which was consistent with previous digital program studies [7,38,39]. Previous research has shown that the digitalization of rehabilitation can improve the professional-patient relationship in meeting the individual needs of the patient [1–4,38,39]. Web-based counseling could encourage patients to find self-awareness for managing a positive life attitude by giving them time to ask questions and listen critically [32] and helping them to ask what is important in life [25]. Web-based support and counseling with individual feedback enhances patients' own potential to achieve goals [5,10]. Receiving individual, positive and realistic feedback is also a key part of the disease acceptance process in the remote context [16,17]. The use of technology as part of the trust-building process supports and enables lifestyle change and related thinking and experience sharing, goal tracking, and information seeking, which are also important coping strategies [24].

## Finding hope and forgetting about the disease

Cardiac patients described their reactions to their illness and how finding their self-awareness and forgetting about thoughts about the illness helped them maintain a positive life attitude. In the trust-building process, they found hope to live and forget about the disease in their daily lives, which is consistent with Schaufel's (2011) study results [18]. There are similarities between the awareness process expressed in this study and the processes described by the psychological models that focus on reorientation, the life situation, acceptance to form a new or restored sense of self-concept, renewed values and the identification of new meanings in life [22–24]. Previous theories and studies have shown that patients need to have emotional management support that enables them to deal with their own thoughts and feelings toward different coping strategies [17,19,21,32]. One interesting finding of these studies is that patients' own perceptions of their ability to cope with illness and change guide their thinking processes and motivation. This finding can help multidisciplinary rehabilitation staff understand how patients cope to deepen their understanding of the disease and how to support patients in the rehabilitation process.

Finally, this study is limited because it examined the meanings of the cardiac rehabilitation process for patients in only one rehabilitation setting and because the data collection included only group interviews. Group interviews have strengths and weaknesses compared to one-on-one interviews. However, the patients spoke freely in a familiar peer group. On the other hand, the strengths of the research include the careful and transparent analysis process. The results

will be of interest to scholars in this field, and they highlight the importance of the usefulness of digital rehabilitation in its implementation.

## Conclusion

The purpose of the current study was to provide an understanding of cardiac patients' experiences of the meaning of cardiac digital rehabilitation for their lives. This study showed that rehabilitation includes a complex process of trust-building, with interaction between cognition, emotion, acceptance and support processes. Within this complex trust-building complex process, multiple categories of trust-building were observed, including feeling that one has hit rock bottom, facing and coping in a crosscurrent, understanding together as a peer group, moving toward a healthier lifestyle with technology, and finding self-awareness. Involvement in the peer support group was particularly important for patients. It can be concluded that emotion management counseling and a face-to-face or digital peer format should be available to patients in digital rehabilitation for them to build trust.

## Acknowledgments

The authors would like to thank all cardiac patients for their helpful collaboration. The authors would like to thank Anita Malinen from the University of Jyväskylä, Finland; and Kamran Namdar from the University of Mälardalen, Sweden and the following research group members: Mika Pekkonen from the rehabilitation center in Peurunka, Finland; Jarkko Honkonen from the Social Insurance Institution, Finland.

## Author Contributions

**Conceptualization:** Marjo-Riitta Anttila.

**Formal analysis:** Marjo-Riitta Anttila, Tuulikki Sjögren.

**Funding acquisition:** Tuulikki Sjögren.

**Investigation:** Marjo-Riitta Anttila, Anne Söderlund, Tuulikki Sjögren.

**Methodology:** Marjo-Riitta Anttila, Anne Söderlund, Tuulikki Sjögren.

**Project administration:** Tuulikki Sjögren.

**Software:** Marjo-Riitta Anttila.

**Supervision:** Anne Söderlund, Tuulikki Sjögren.

**Visualization:** Marjo-Riitta Anttila, Anne Söderlund.

**Writing – original draft:** Marjo-Riitta Anttila.

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
