## [Decision Letter · Decision Letter 0]

25 Sep 2020

PONE-D-20-20409

Patients' experiences of the complex trust-building process within digital cardiac rehabilitation

PLOS ONE

Dear Dr. Anttila,

Thank you for submitting your manuscript to PLOS ONE. After careful consideration, we feel that it has merit but does not fully meet PLOS ONE’s publication criteria as it currently stands. Therefore, we invite you to submit a revised version of the manuscript that addresses the points raised during the review process.

We look forward to receiving your revised manuscript.

Kind regards,

Celena Scheede-Bergdahl

Academic Editor

PLOS ONE

Additional Editor Comments:

Reviewer's comments verified by Academic Editor. AE in agreement.

Journal Requirements:

2. Please include additional information regarding the interview guides used in the study and ensure that you have provided sufficient details that others could replicate the analyses. For instance, if you developed a questionnaire as part of this study and it is not under a copyright more restrictive than CC-BY, please include a copy, in both the original language and English, as Supporting Information.

Reviewers' comments:

Reviewer's Responses to Questions

**Comments to the Author**

1. Is the manuscript technically sound, and do the data support the conclusions?

Reviewer #1: Yes

2. Has the statistical analysis been performed appropriately and rigorously? 

Reviewer #1: Yes

3. Have the authors made all data underlying the findings in their manuscript fully available?

Reviewer #1: Yes

4. Is the manuscript presented in an intelligible fashion and written in standard English?

Reviewer #1: Yes

5. Review Comments to the Author

Reviewer #1: Interesting study. Generally well-written.

The manuscript could be improved by adding a demographics tables (gender, age, race, ethnicity, SES, marital status, employment status, age at onset ect).

Please include additional information on family history, medications taken and any comorbidities.

Also, include information re: access to digital technology.

Inclusion and exclusion criteria needs additional description

"22 males and 8 females, who had undergone coronary angioplasty (25/30) or bypass

103 surgery (4/30) approximately 3 to 12 months prior to rehabilitation".

6. PLOS authors have the option to publish the peer review history of their article (what does this mean?). If published, this will include your full peer review and any attached files.

Reviewer #1: No

---

## [Author Response · Author response to Decision Letter 0]

8 Feb 2021

Response to Reviewer PLOS ONE

Dear Academic Editor Celena Scheede-Bergdahl and reviewer

We want to thank you for giving us the opportunity to revise the manuscript entitled (PONE-D-20-20409) “Patients' experiences of the complex trust-building process within digital cardiac rehabilitation". The manuscript has been revised based on the comments of the reviewer and academic editor. We respond to academic editor and reviewer comments and submit one revised manuscript. We want to thank the reviewer for constructive and important comments and these comments helping us to improve our manuscript. Below we have listed all comments with a point-by-point response. Two versions of the revised manuscript are submitted: a clean final copy and a version with all changes marked. 

Academic Editor comments

Journal Requirements:

When submitting your revision, we need you to address these additional requirements

1. Please ensure that your manuscript meets PLOS ONE's style requirements

Comments to the Editor: We has been taken account PLOS ONE's style requirements.

2. Please include additional information regarding the interview guides used in the study and ensure that you have provided sufficient details that others could replicate the analyses. For instance, if you developed a questionnaire as part of this study and it is not under a copyright more restrictive than CC-BY, please include a copy, in both the original language and English, as Supporting Information.

Comments to the Editor: Interview guide information in the data collection and analyzing section as follows:

 Each focus group interview included 5-9 patients and lasted an average of 30-90 minutes. The participants were encouraged to speak openly about their experiences of the rehabilitation process. The interviewer was free to vary the questions and their content and to allow for group discussion. Example questions, are as follows: “What were your meaningful experiences with rehabilitation process?”; “How was the peer group meaningful to you?”; and “What are your experiences of how technology has been a help to you in coping with daily life and lifestyle changes?” The interviewer asked follow-up questions as needed. The interviewer observed the group dynamics and ensured that everyone had an opportunity to speak. Occasionally, there were people in the focus group who talked more and guided the session, but the interviewer provided a balanced discussion by asking the quieter participants supplementary questions.

The study analyzed patients’ experiences of the rehabilitation process in accordance with principles of Glaser’s` grounded theory (open, selective coding), which guided the constant comparative model (See: Theoretical sensitivity: advances in the methodology of grounded theory. Mill Valley (Calif.): Sociology Press; 1978). We have described the analysis process in the text and figure 1. We added registered information to article text: The study is a part of a remote technology in cardiac rehabilitation study registered at the ISRCTN Registry [ISRCTN61225589]. 

3. Please provide additional details regarding participant consent. In the ethics statement in the Methods and online submission information, please ensure that you have specified (1) whether consent was informed and (2) what type you obtained (for instance, written or verbal, and if verbal, how it was documented and witnessed). If your study included minors, state whether you obtained consent from parents or guardians. If the need for consent was waived by the ethics committee, please include this information 

Comments to the Editor: 

Participant consent Information has been added in the methods section of the manuscript, as follows: Patients were informed verbal and written of the study prior to rehabilitation that participation was voluntary and that they could withdraw from the study at any time. Patients signed informed consent forms to participate in the study. Patients were also assured that whether they wanted to participate in the study or not, it would not have any negative impact on the rehabilitation process. (Page 5, setting section, line 118-122)

The information is also in the ethical statement, The Ethics Committee of the Central Finland Health Care (2015).

No minors participated in the study

4. Please clarify the sources of funding (financial or material support) for your study. List the grants or organizations that supported your study, including funding received from your institution.

Comments to the Editor: Here is more information about the sources of funding, which were University of Jyväskylä and Social insurance institution of Finland (Kela).

Comments to the Editor: We designed the research-project, data collection and analysis and applied for funding from the Social Insurance Institution (Kela). The funder approved the research plan and provided funding to the grant for intervention, financial support for a research project. In Finland Kela carries out research on the benefit systems it operates, Kela’s internal operations, general health security and rehabilitation (https://www.kela.fi/web/en/research-projects). 

The University of Jyväskylä is employer by researchers, Tuulikki Sjögren and Marjo-Riitta Anttila (salary).

Comments to the Editor: 

d.If you did not receive any funding for this study, please state: “The authors received no specific funding for this work.”

Comments to the Editor: The authors received no specific funding for this work (article). The funders (Kela) had no role (no salary) in decision to publish, or preparation of the manuscript by any authors.

1. Reviewer

Reviewer #1: Interesting study. Generally well-written. The manuscript could be improved by adding a demographics tables (gender, age, race, ethnicity, SES, marital status, employment status, age at onset ect). Please include additional information on family history, medications taken and any comorbidities. Also, include information re: access to digital technology. Inclusion and exclusion criteria needs additional description "22 males and 8 females, who had undergone coronary angioplasty (25/30) or bypass 103 surgery (4/30) approximately 3 to 12 months prior to rehabilitation".

Author response

We thank the reviewer #1 for these comments and comments related to the research topic have been taken into account as follows.

1.2 Reviewer comments to the author

The manuscript could be improved by adding a demographics tables (gender, age, race, ethnicity, SES, marital status, employment status, age at onset ect).

Author response/action:

We agree with the reviewer. We were improved the manuscript by adding a demographics table (gender, mean age, marital status, education, employment status, operation, time of operation) (Page 6, setting section, line 126-139).

1.3 Reviewer comments to the author

Please include additional information on family history, medications taken and any comorbidities. 

Author response/action:

We agree with the reviewer. We appreciate the concerns of the reviewer and it would have been interesting to have this information, but unfortunately, we didn`t collected additional information (family history, medications taken and any comorbidities) during this qualitative study.

 1.4 Reviewer comments to the author

Also, include information re: access to digital technology. Inclusion and exclusion criteria needs additional description "22 males and 8 females, who had undergone coronary angioplasty (25/30) or bypass 103 surgery (4/30) approximately 3 to 12 months prior to rehabilitation".

Author response/action:

We agree with the reviewer. We have restructured the sections and added to the text inclusion and exclusion criteria as follows (Page 5-6, setting section, line 108-118):

Semistructured interviews were conducted with cardiac patients (mean ages 55.46), including 22 males and 8 females, who had undergone coronary angioplasty (25/30) or bypass surgery (4/30) approximately 3 to 12 months prior to rehabilitation is presented. In table 1 demographic information and operation description at baseline. The participants were recruited from rehabilitation groups (n=4) that utilized digital health tools for traditional cardiac rehabilitation in Finland. The participants were recruited among adult patients with coronary artery disease (CAD) and attended the cardiac rehabilitation courses that utilized digital health tools for traditional cardiac rehabilitation (Peurunka rehabilitation centre, Finland). The inclusion criteria for participation were adults (18 years or older) who had cardiovascular risk factors or angina pectoris with physical working capacity limitation or myocardial infarction or coronary artery bypass graft surgery (CABG) or coronary angioplasty (PTCA). Participants were excluded if they had musculoskeletal problems, cognitive or memory impairment, or if they were unable to use independently computer and remote technology application. Participation in the study intervention also required the participant to have internet at home or be able to use a computer of family members or friends. Participants in the study received an activity tracker accelerometer (Fitbit Charge HR®, USA) and remote coaching tool software (m-coach Movendos, Finland), which was free to download on computers, tablets and smart phones. 

Kind regards,

Marjo-Riitta Anttila

Corresponding Author:

Marjo-Riitta Anttila, MSc

Faculty of Sport and Health Sciences 

University of Jyväskylä

P. O. Box 35

Jyväskylä, FI-40014

Finland

Phone:+358408054648

Fax: 358142602011

Email: marjo-riitta.m-r.anttila@jyu.fi

---

## [Editor Report · Decision Letter 1]

18 Feb 2021

Patients' experiences of the complex trust-building process within digital cardiac rehabilitation

PONE-D-20-20409R1

Dear Dr. Marjo-Riitta Anttila,

We’re pleased to inform you that your manuscript has been judged scientifically suitable for publication and will be formally accepted for publication once it meets all outstanding technical requirements.

Kind regards,

Celena Scheede-Bergdahl

Academic Editor

PLOS ONE

Additional Editor Comments (optional):

Comments sufficiently addressed.
---

## [Editor Report · Acceptance letter]

25 Feb 2021

PONE-D-20-20409R1 

Patients' experiences of the complex trust-building process within digital cardiac rehabilitation 

Dear Dr. Anttila:

I'm pleased to inform you that your manuscript has been deemed suitable for publication in PLOS ONE. Congratulations! Your manuscript is now with our production department. 

Kind regards, 

on behalf of

Dr. Celena Scheede-Bergdahl 

Academic Editor

PLOS ONE